# MomentumRNN: Integrating Momentum into Recurrent Neural Networks

**Tan M. Nguyen**
Department of ECE
Rice University, Houston, USA

**Richard G. Baraniuk**
Department of ECE
Rice University, Houston, USA

**Andrea L. Bertozzi**
Department of Mathematics
University of California, Los Angeles

**Stanley J. Osher**
Department of Mathematics
University of California, Los Angeles

**Bao Wang** *
Department of Mathematics
Scientific Computing and Imaging (SCI) Institute
University of Utah, Salt Lake City, UT, USA

## Abstract

Designing deep neural networks is an art that often involves an expensive search over candidate architectures. To overcome this for recurrent neural nets (RNNs), we establish a connection between the hidden state dynamics in an RNN and gradient descent (GD). We then integrate momentum into this framework and propose a new family of RNNs, called *MomentumRNNs*. We theoretically prove and numerically demonstrate that MomentumRNNs alleviate the vanishing gradient issue in training RNNs. We study the momentum long-short term memory (MomentumLSTM) and verify its advantages in convergence speed and accuracy over its LSTM counterpart across a variety of benchmarks. We also demonstrate that MomentumRNN is applicable to many types of recurrent cells, including those in the state-of-the-art orthogonal RNNs. Finally, we show that other advanced momentum-based optimization methods, such as Adam and Nesterov accelerated gradients with a restart, can be easily incorporated into the MomentumRNN framework for designing new recurrent cells with even better performance.

## 1 Introduction

Mathematically principled recurrent neural nets (RNNs) facilitate the network design process and reduce the cost of searching over many candidate architectures. A particular advancement in RNNs is the long short-term memory (LSTM) model [24] which has achieved state-of-the-art results in many applications, including speech recognition [15], acoustic modeling [53, 51], and language modeling [46]. There have been many efforts in improving LSTM: [19] introduces a forget gate into the original LSTM cell, which can forget information selectively; [18] further adds peephole connections to the LSTM cell to inspect its current internal states[17]; to reduce the computational cost, a gated recurrent unit (GRU) [11] uses a single update gate to replace the forget and input gates in LSTM. Phased LSTM [42] adds a new time gate to the LSTM cell and achieves faster convergence than the regular LSTM on learning long sequences. In addition, [52] and [50] introduce a biological cell state and working memory into LSTM, respectively. Nevertheless, most of RNNs, including LSTMs, are biologically informed or even ad-hoc instead of being guided by mathematical principles.

## 1.1 Recap on RNNs and LSTM

Recurrent cells are the building blocks of RNNs. A recurrent cell employs a cyclic connection to update the current hidden state ($h_t$) using the past hidden state ($h_{t-1}$) and the current input data ($x_t$) [14]; the dependence of $h_t$ on $h_{t-1}$ and $x_t$ in a recurrent cell can be written as

$$h_t = \sigma(\mathbf{U}h_{t-1} + \mathbf{W}x_t + b), \ x_t \in \mathbb{R}^d, \ \text{and} \ h_{t-1}, h_t \in \mathbb{R}^h, \ t = 1, 2, \cdots, T, \qquad (1)$$

where $\mathbf{U} \in \mathbb{R}^{h \times h}, \mathbf{W} \in \mathbb{R}^{h \times d}$, and $b \in \mathbb{R}^h$ are trainable parameters; $\sigma(\cdot)$ is a nonlinear activation function, e.g., sigmoid or hyperbolic tangent. Error backpropagation through time is used to train RNN, but it tends to result in exploding or vanishing gradients [4]. Thus RNNs may fail to learn long term dependencies. Several approaches exist to improve RNNs' performance, including enforcing unitary weight matrices [1, 62, 25, 60, 38, 22], leveraging LSTM cells, and others [35, 30].

LSTM cells augment the recurrent cell with "gates" [24] and can be formulated as

$$
\begin{aligned}
i_t &= \sigma(\mathbf{U}_{ih}h_{t-1} + \mathbf{W}_{ix}x_t + b_i), &&(i_t : \text{input gate}) \\
\widetilde{c}_t &= \tanh\left(\mathbf{U}_{\widetilde{c}h}h_{t-1} + \mathbf{W}_{\widetilde{c}x}x_t + b_{\widetilde{c}}\right), &&(\widetilde{c}_t : \text{cell input}) \\
c_t &= c_{t-1} + i_t \odot \widetilde{c}_t, &&(c_t : \text{cell state}) \qquad (2) \\
o_t &= \sigma(\mathbf{U}_{oh}h_{t-1} + \mathbf{W}_{ox}x_t + b_o), &&(o_t : \text{output gate}) \\
h_t &= o_t \odot \tanh c_t, &&(h_t : \text{hidden state})
\end{aligned}
$$

where $\mathbf{U}_* \in \mathbb{R}^{h \times h}, \mathbf{W}_* \in \mathbb{R}^{h \times d}$, and $b_* \in \mathbb{R}^h$ are learnable parameters, and $\odot$ denotes the Hadamard product. The input gate decides what new information to be stored in the cell state, and the output gate decides what information to output based on the cell state value. The gating mechanism in LSTMs can lead to the issue of saturation [59, 8].

## 1.2 Our Contributions

In this paper, we develop a gradient descent (GD) analogy of the recurrent cell. In particular, the hidden state update in a recurrent cell is associated with a gradient descent step towards the optimal representation of the hidden state. We then propose to integrate momentum that used for accelerating gradient dynamics into the recurrent cell, which results in the momentum cell. At the core of the momentum cell is the use of momentum to accelerate the hidden state learning in RNNs. The architectures of the standard recurrent cell and our momentum cell are illustrated in Fig. 1. We provide the design principle and detailed derivation of the momentum cell in Sections 2.2 and 2.4. We call the RNN that consists of momentum cells the MomentumRNN. The major advantages of MomentumRNN are fourfold:

- MomentumRNN can alleviate the vanishing gradient problem in training RNN.
- MomentumRNN accelerates training and improves the test accuracy of the baseline RNN.
- MomentumRNN is universally applicable to many existing RNNs. It can be easily implemented by changing a few lines of the baseline RNN code.
- MomentumRNN is principled with theoretical guarantees provided by the momentum-accelerated dynamical system for optimization and sampling. The design principle can be generalized to other advanced momentum-based optimization methods, including Adam [28] and Nesterov accelerated gradients with a restart [44, 61].

## 1.3 Related Work

**Dynamical system viewpoint of RNNs.** Leveraging the theory of dynamical system to improve RNNs has been an interesting research direction: [31] proposes a gated RNN, which is principled by non-chaotical dynamical systems and achieves comparable performance to GRUs and LSTMs. [57] proposes a weight initialization strategy inspired by dynamical system theory, which helps the training of RNNs with ReLU nonlinearity. Other RNN algorithms derived from the dynamical system theories include [45, 9, 10, 26]. Our work is the first that directly integrates momentum into an RNN to accelerate the underlying dynamics and improve the model's performance.

**Momentum in Optimization and Sampling.** Momentum has been a popular technique for accelerating (stochastic) gradient-based optimization [49, 20, 55, 28, 3, 48] and sampling algorithms [13, 41] A particularly interesting momentum is the iteration-dependent one in NAG [44, 43, 2], which has a significantly better convergence rate than constant momentum for convex optimization. The stochastic gradient NAG that employs a scheduled restart can also be used to accelerate DNN training with better accuracy and faster convergence [61].

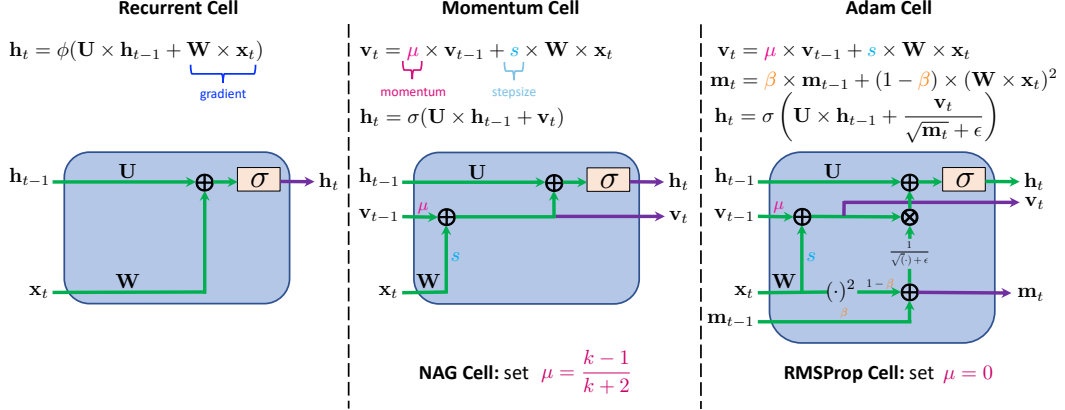

Figure 1: Illustration of the recurrent cell (left), Momentum/NAG cell (middle), and Adam/RMSProp cell (right). We draw a connection between the dynamics of hidden states in the recurrent cell and GD. We then introduce momentum to recurrent cell as an analogy of the momentum accelerated GD.

**Momentum in DNNs.** Momentum has also been used in designing DNN architectures. [21] develops momentum contrast as a way of building large and consistent dictionaries for unsupervised learning with contrastive loss. At the core of this approach is a momentum-based moving average of the queue encoder. Many DNN-based algorithms for sparse coding are designed by unfolding the classical optimization algorithms, e.g., FISTA [2], in which momentum can be used in the underpinning optimizer [56, 7, 36, 27, 40].

## 1.4 Notation

We denote scalars by lower or upper case letters; vectors and matrices by lower and upper case bold face letters, respectively. For a vector $\boldsymbol{x} = (x_1, \cdots, x_d)^T \in \mathbb{R}^d$, we use $\|\boldsymbol{x}\| = (\sum_{i=1}^d |x_i|^2)^{1/2}$ to denote its $\ell_2$ norm. For a matrix $\mathbf{A}$, we use $\mathbf{A}^{\mathrm{T}}$ (T in roman type) and $\mathbf{A}^{-1}$ to denote its transpose and inverse, respectively. Also, we denote the spectral norm of $\mathbf{A}$ as $\|\mathbf{A}\|$. We denote the $d$-dimensional standard Gaussian as $\mathcal{N}(\mathbf{0}, \mathbf{I}_{d \times d})$, where $\mathbf{0}$ is the $d$-dimensional zero-vector and $\mathbf{I}_{d \times d}$ is an identity matrix. For a function $\phi(\boldsymbol{x}) : \mathbb{R}^d \to \mathbb{R}$, we denote $\phi^{-1}(\boldsymbol{x})$ as its inverse and $\nabla \phi(\boldsymbol{x})$ as its gradient.

# 2 Momentum RNNs

## 2.1 Background: Momentum Acceleration for Gradient Based Optimization and Sampling

Momentum has been successfully used to accelerate the gradient-based algorithms for optimization and sampling. In optimization, we aim to find a stationary point of a given function $f(\boldsymbol{x}), \boldsymbol{x} \in \mathbb{R}^d$. Starting from $\boldsymbol{x}_0 \in \mathbb{R}^d$, GD iterates as $\boldsymbol{x}_t = \boldsymbol{x}_{t-1} - s\nabla f(\boldsymbol{x}_t)$ with $s > 0$ being the step size. This can be significantly accelerated by using the momentum [55], which results in

$$\boldsymbol{p}_0 = \boldsymbol{x}_0; \ \boldsymbol{p}_t = \mu\boldsymbol{p}_{t-1} + s\nabla f(\boldsymbol{x}_t); \ \boldsymbol{x}_t = \boldsymbol{x}_{t-1} - \boldsymbol{p}_t, \ t \geq 1, \tag{3}$$

where $\mu \geq 0$ is the momentum constant. In sampling, Langevin Monte Carlo (LMC) [12] is used to sample from the distribution $\pi \propto \exp\{-f(\boldsymbol{x})\}$, where $\exp\{-f(\boldsymbol{x})\}$ is the probability distribution function. The update at each iteration is given by

$$\boldsymbol{x}_t = \boldsymbol{x}_{t-1} - s\nabla f(\boldsymbol{x}_t) + \sqrt{2s}\boldsymbol{\epsilon}_t, \ s \geq 0, \ t \geq 1, \ \boldsymbol{\epsilon}_t \sim \mathcal{N}(\mathbf{0}, \mathbf{I}_{d \times d}). \tag{4}$$

We can also use momentum to accelerate LMC, which results in the following Hamiltonian Monte Carlo (HMC) update [12]:

$$\boldsymbol{p}_0 = \boldsymbol{x}_0; \ \boldsymbol{p}_t = \boldsymbol{p}_{t-1} - \gamma s\boldsymbol{p}_{t-1} - s\eta\nabla f(\boldsymbol{x}_{t-1}) + \sqrt{2\gamma s\eta}\boldsymbol{\epsilon}_t; \ \boldsymbol{x}_t = \boldsymbol{x}_{t-1} + s\boldsymbol{p}_t, \ t \geq 1, \tag{5}$$

where $\boldsymbol{\epsilon}_t \sim \mathcal{N}(\mathbf{0}, \mathbf{I}_{d \times d})$ while $\gamma, \eta, s > 0$ are the friction parameter, inverse mass, and step size, resp.

## 2.2 Gradient Descent Analogy for RNN and MomentumRNN

Now, we are going to establish a connection between RNN and GD, and further leverage momentum to improve RNNs. Let $\widetilde{\mathbf{W}} = [\mathbf{W}, \boldsymbol{b}]$ and $\widetilde{\boldsymbol{x}}_t = [\boldsymbol{x}_t, 1]^T$ in (1), then we have $\boldsymbol{h}_t = \sigma(\mathbf{U}\boldsymbol{h}_{t-1} + \widetilde{\mathbf{W}}\widetilde{\boldsymbol{x}}_t)$. For the ease of notation, without ambiguity we denote $\mathbf{W} := \widetilde{\mathbf{W}}$ and $\boldsymbol{x}_t := \widetilde{\boldsymbol{x}}_t$. Then the recurrent cell can be reformulated as

$$\boldsymbol{h}_t = \sigma(\mathbf{U}\boldsymbol{h}_{t-1} + \mathbf{W}\boldsymbol{x}_t). \tag{6}$$

Moreover, let $\phi(\cdot) := \sigma(\mathbf{U}(\cdot))$ and $\boldsymbol{u}_t := \mathbf{U}^{-1}\mathbf{W}\boldsymbol{x}_t$, we can rewrite (6) as

$$\boldsymbol{h}_t = \phi(\boldsymbol{h}_{t-1} + \boldsymbol{u}_t). \tag{7}$$

If we regard $-\boldsymbol{u}_t$ as the "gradient" at the $t$-th iteration, then we can consider (7) as the dynamical system which updates the hidden state by the gradient and then transforms the updated hidden state by the nonlinear activation function $\phi$. We propose the following accelerated dynamical system to accelerate the dynamics of (7), which is principled by the accelerated gradient descent theory (see subsection 2.1):

$$\boldsymbol{p}_t = \mu\boldsymbol{p}_{t-1} - s\boldsymbol{u}_t; \ \ \boldsymbol{h}_t = \phi(\boldsymbol{h}_{t-1} - \boldsymbol{p}_t), \tag{8}$$

where $\mu \geq 0, s > 0$ are two hyperparameters, which are the analogies of the momentum coefficient and step size in the momentum-accelerated GD, respectively. Let $\mathbf{v}_t := -\mathbf{U}\boldsymbol{p}_t$, we arrive at the following dynamical system:

$$\mathbf{v}_t = \mu\mathbf{v}_{t-1} + s\mathbf{W}\boldsymbol{x}_t; \ \ \boldsymbol{h}_t = \sigma(\mathbf{U}\boldsymbol{h}_{t-1} + \mathbf{v}_t). \tag{9}$$

The architecture of the momentum cell that corresponds to the dynamical system (9) is plotted in Fig. 1 (middle). Compared with the recurrent cell, the momentum cell introduces an auxiliary momentum state in each update and scales the dynamical system with two positive hyperparameters $\mu$ and $s$.

**Remark 1** *Different parameterizations of (8) can result in different momentum cell architectures. For instance, if we let $\mathbf{v}_t = -\boldsymbol{p}_t$, we end up with the following dynamical system:*

$$\mathbf{v}_t = \mu\mathbf{v}_{t-1} + s\widehat{\mathbf{W}}\boldsymbol{x}_t; \ \ \boldsymbol{h}_t = \sigma(\mathbf{U}\boldsymbol{h}_{t-1} + \mathbf{U}\mathbf{v}_t), \tag{10}$$

*where $\widehat{\mathbf{W}} := \mathbf{U}^{-1}\mathbf{W}$ is the trainable weight matrix. Even though (9) and (10) are mathematically equivalent, the training procedure might cause the MomentumRNNs that are derived from different parameterizations to have different performances.*

**Remark 2** *We put the nonlinear activation in the second equation of (8) to ensure that the value of $\boldsymbol{h}_t$ is in the same range as the original recurrent cell.*

**Remark 3** *The derivation above also applies to the dynamical systems in the LSTM cells, and we can design the MomentumLSTM in the same way as designing the MomentumRNN.*

### 2.3   Analysis of the Vanishing Gradient Issue: Momentum Cell vs. Recurrent Cell

Let $\boldsymbol{h}_T$ and $\boldsymbol{h}_t$ be the state vectors at the time step $T$ and $t$, respectively, and we suppose $T \gg t$. Furthermore, assume that $\mathcal{L}$ is the objective to minimize, then

$$\frac{\partial\mathcal{L}}{\partial\boldsymbol{h}_t} = \frac{\partial\mathcal{L}}{\partial\boldsymbol{h}_T} \cdot \frac{\partial\boldsymbol{h}_T}{\partial\boldsymbol{h}_t} = \frac{\partial\mathcal{L}}{\partial\boldsymbol{h}_T} \cdot \prod_{k=t}^{T-1}\frac{\partial\boldsymbol{h}_{k+1}}{\partial\boldsymbol{h}_k} = \frac{\partial\mathcal{L}}{\partial\boldsymbol{h}_T} \cdot \prod_{k=t}^{T-1}(\mathbf{D}_k\mathbf{U}^\mathrm{T}), \tag{11}$$

where $\mathbf{U}^\mathrm{T}$ is the transpose of $\mathbf{U}$ and $\mathbf{D}_k = \mathrm{diag}(\sigma'(\mathbf{U}\boldsymbol{h}_k + \mathbf{W}\boldsymbol{x}_{k+1}))$ is a diagonal matrix with $\sigma'(\mathbf{U}\boldsymbol{h}_k + \mathbf{W}\boldsymbol{x}_{k+1})$ being its diagonal entries. $\|\prod_{k=t}^{T-1}(\mathbf{D}_k\mathbf{U}^\mathrm{T})\|_2$ tends to either vanish or explode [4]. We can use regularization or gradient clipping to mitigate the exploding gradient, leaving vanishing gradient as the major obstacle to training RNN to learn long-term dependency [47]. We can rewrite (9) as

$$\boldsymbol{h}_t = \sigma\left(\mathbf{U}(\boldsymbol{h}_{t-1} - \mu\boldsymbol{h}_{t-2}) + \mu\sigma^{-1}(\boldsymbol{h}_{t-1}) + s\mathbf{W}\boldsymbol{x}_t\right), \tag{12}$$

where $\sigma^{-1}(\cdot)$ is the inverse function of $\sigma(\cdot)$. We compute $\partial\mathcal{L}/\partial\boldsymbol{h}_t$ as follows

$$\frac{\partial\mathcal{L}}{\partial\boldsymbol{h}_t} = \frac{\partial\mathcal{L}}{\partial\boldsymbol{h}_T} \cdot \frac{\partial\boldsymbol{h}_T}{\partial\boldsymbol{h}_t} = \frac{\partial\mathcal{L}}{\partial\boldsymbol{h}_T} \cdot \prod_{k=t}^{T-1}\frac{\partial\boldsymbol{h}_{k+1}}{\partial\boldsymbol{h}_k} = \frac{\partial\mathcal{L}}{\partial\boldsymbol{h}_T} \cdot \prod_{k=t}^{T-1}\widehat{\mathbf{D}}_k[\mathbf{U}^\mathrm{T} + \mu\boldsymbol{\Sigma}_k], \tag{13}$$

where $\widehat{\mathbf{D}}_k = \mathrm{diag}(\sigma'(\mathbf{U}(\boldsymbol{h}_k - \mu\boldsymbol{h}_{k-1}) + \mu\sigma^{-1}(\boldsymbol{h}_k) + s\mathbf{W}\boldsymbol{x}_{k+1}))$ and $\boldsymbol{\Sigma} = \mathrm{diag}((\sigma^{-1})'(\boldsymbol{h}_k))$. For mostly used $\sigma$, e.g., sigmoid and tanh, $(\sigma^{-1}(\cdot))' > 1$ and $\mu\boldsymbol{\Sigma}_k$ dominates $\mathbf{U}^\mathrm{T}$.[2] Therefore, with an appropriate choice of $\mu$, the momentum cell can alleviate vanishing gradient and accelerate training.

We empirically corroborate that momentum cells can alleviate vanishing gradients by training a MomentumRNN and its corresponding RNN on the PMNIST classification task and plot $\|\partial\mathcal{L}/\partial\boldsymbol{h}_t\|_2$ for each time step $t$. Figure 2 confirms that unlike in RNN, the gradients in MomentumRNN do not vanish. More details on this experiment are provided in the Appendix A.

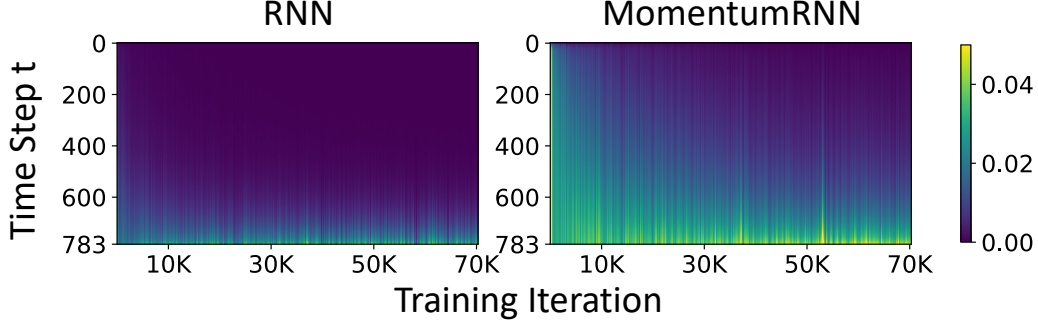

Figure 2: $\ell_2$ norm of the gradients of the loss $\mathcal{L}$ w.r.t. the state vector $\boldsymbol{h}_t$ at each time step $t$ for RNN (left) and MomentumRNN (right). MomentumRNN does not suffer from vanishing gradients.

### 2.4 Beyond MomentumRNN: NAG and Adam Principled Recurrent Neural Nets

There are several other advanced formalisms of momentum existing in optimization, which can be leveraged for RNN architecture design. In this subsection, we present two additional variants of MomentumRNN that are derived from the Nesterov accelerated gradient (NAG)-style momentum with restart [44, 61] and Adam [28].

**NAG Principled RNNs.** The momentum-accelerated GD can be further accelerated by replacing the constant momentum coefficient $\mu$ in (9) with the NAG-style momentum, i.e. setting $\mu$ to $(t-1)/(t+2)$ at the $t$-th iteration. Furthermore, we can accelerate NAG by resetting the momentum to 0 after every $F$ iterations, i.e. $\mu = (t \mod F)/((t \mod F) + 3)$, which is the NAG-style momentum with a scheduled restart of the appropriately selected frequency $F$ [61]. For convex optimization, NAG has a convergence rate $O(1/t^2)$, which is significantly faster than GD or GD with constant momentum whose convergence rate is $O(1/t)$. Scheduled restart not only accelerates NAG to a linear convergence rate $O(\alpha^{-t})(0 < \alpha < 1)$ under mild extra assumptions but also stabilizes the NAG iteration [61]. We call the MomentumRNN with the NAG-style momentum and scheduled restart momentum the NAG-based RNN and the scheduled restart RNN (SRRNN), respectively.

**Adam Principled RNNs.** Adam [28] leverages the moving average of historical gradients and entry-wise squared gradients to accelerate the stochastic gradient dynamics. We use Adam to accelerate (7) and end up with the following iteration

$$\boldsymbol{p}_t = \mu\boldsymbol{p}_{t-1} + (1-\mu)\boldsymbol{u}_t; \; \boldsymbol{m}_t = \beta\boldsymbol{m}_{t-1} + (1-\beta)\boldsymbol{u}_t \odot \boldsymbol{u}_t; \; \boldsymbol{h}_t = \phi(\boldsymbol{h}_{t-1} - s\frac{\boldsymbol{p}_t}{\sqrt{\boldsymbol{r}_t} + \epsilon}), \quad (14)$$

where $\mu, s, \beta > 0$ are hyperparameters, $\epsilon$ is a small constant and chosen to be $10^{-8}$ by default, and $\odot/\sqrt{\cdot}$ denotes the entrywise product/square root[3]. Again, let $\mathbf{v}_t = -\mathbf{U}\boldsymbol{p}_t$, we rewrite (14) as follows

$$\mathbf{v}_t = \mu\mathbf{v}_{t-1} + (1-\mu)\mathbf{W}\boldsymbol{x}_t; \; \boldsymbol{m}_t = \beta\boldsymbol{m}_{t-1} + (1-\beta)\boldsymbol{u}_t \odot \boldsymbol{u}_t; \; \boldsymbol{h}_t = \sigma(\mathbf{U}\boldsymbol{h}_{t-1} + s\frac{\mathbf{v}_t}{\sqrt{\boldsymbol{m}_t} + \epsilon}).$$

As before, here $\boldsymbol{u}_t := \mathbf{U}^{-1}\mathbf{W}\boldsymbol{x}_t$. Computing $\mathbf{U}^{-1}$ is expensive. Our experiments suggest that replacing $\boldsymbol{u}_t \odot \boldsymbol{u}_t$ by $\mathbf{W}\boldsymbol{x}_t \odot \mathbf{W}\boldsymbol{x}_t$ is sufficient and more efficient to compute. In our implementation, we also relax $\mathbf{v}_t = \mu\mathbf{v}_{t-1} + (1-\mu)\mathbf{W}\boldsymbol{x}_t$ to $\mathbf{v}_t = \mu\mathbf{v}_{t-1} + s\mathbf{W}\boldsymbol{x}_t$ that follows the momentum in the MomentumRNN (9) for better performance. Therefore, we propose the *AdamRNN* that is given by

$$\mathbf{v}_t = \mu\mathbf{v}_{t-1} + s\mathbf{W}\boldsymbol{x}_t; \; \boldsymbol{m}_t = \beta\boldsymbol{m}_{t-1} + (1-\beta)(\mathbf{W}\boldsymbol{x}_t \odot \mathbf{W}\boldsymbol{x}_t); \; \boldsymbol{h}_t = \sigma(\mathbf{U}\boldsymbol{h}_{t-1} + \frac{\mathbf{v}_t}{\sqrt{\boldsymbol{m}_t} + \epsilon}).$$
$$(15)$$

In AdamRNN, if $\mu$ is set to 0, we achieve another new RNN, which obeys the RMSProp gradient update rule [58]. We call this new model the *RMSPropRNN*.

**Remark 4** *Both AdamRNN and RMSPropRNN can also be derived by letting $\mathbf{v}_t = -\boldsymbol{p}_t$ and $\widehat{\mathbf{W}} := \mathbf{U}^{-1}\mathbf{W}$ as in Remark 1. This parameterization yields the following formulation for AdamRNN*

$$\mathbf{v}_t = \mu\mathbf{v}_{t-1} + s\widehat{\mathbf{W}}\boldsymbol{x}_t; \; \boldsymbol{m}_t = \beta\boldsymbol{m}_{t-1} + (1-\beta)(\widehat{\mathbf{W}}\boldsymbol{x}_t \odot \widehat{\mathbf{W}}\boldsymbol{x}_t); \; \boldsymbol{h}_t = \sigma(\mathbf{U}\boldsymbol{h}_{t-1} + \frac{\mathbf{U}\mathbf{v}_t}{\sqrt{\boldsymbol{m}_t} + \epsilon}).$$

*Here, we simply need to learn $\widehat{\mathbf{W}}$ and $\mathbf{U}$ without any relaxation. In contrast, we relaxed $\mathbf{U}^{-1}$ to an identity matrix in (15). Our experiments suggest that both parameterizations yield similar results.*

Table 1: Best test accuracy at the MNIST and PMNIST tasks (%). We use the baseline results reported in [22], [62], [60]. All of our proposed models outperform the baseline LSTM. Among the models using $N = 256$ hidden units, RMSPropLSTM yields the best results in both tasks.

| MODEL | N | # PARAMS | MNIST | PMNIST |
|---|---|---|---|---|
| LSTM | 128 | $\approx 68K$ | 98.70[22],97.30 [60] | 92.00 [22],92.62 [60] |
| LSTM | 256 | $\approx 270K$ | 98.90 [22], 98.50 [62] | 92.29 [22], 92.10 [62] |
| MOMENTUMLSTM | 128 | $\approx 68K$ | **99.04 ± 0.04** | **93.40 ± 0.25** |
| MOMENTUMLSTM | 256 | $\approx 270K$ | **99.08 ± 0.05** | **94.72 ± 0.16** |
| ADAMLSTM | 256 | $\approx 270K$ | 99.09 ± 0.03 | 95.05 ± 0.37 |
| RMSPROPLSTM | 256 | $\approx 270K$ | **99.15 ± 0.06** | **95.38 ± 0.19** |
| SRLSTM | 256 | $\approx 270K$ | 99.01 ± 0.07 | 93.82 ± 1.85 |

# 3 Experimental Results

In this section, we evaluate the effectiveness of our momentum approach in designing RNNs in terms of convergence speed and accuracy. We compare the performance of the MomentumLSTM with the baseline LSTM [24] in the following tasks: 1) the object classification task on pixel-permuted MNIST [32], 2) the speech prediction task on the TIMIT dataset [1, 22, 62, 38, 23], 3) the celebrated copying and adding tasks [24, 1], and 4) the language modeling task on the Penn TreeBank (PTB) dataset [39]. These four tasks are among standard benchmarks to measure the performance of RNNs and their ability to handle long-term dependencies. Also, these tasks cover different data modalities – image, speech, and text data – as well as a variety of model sizes, ranging from thousands to millions of parameters with one (MNIST and TIMIT tasks) or multiple (PTB task) recurrent cells in concatenation. Our experimental results confirm that MomentumLSTM converges faster and yields better test accuracy than the baseline LSTM across tasks and settings. We also discuss the AdamLSTM, RMSPropLSTM, and scheduled restart LSTM (SRLSTM) and show their advantage over MomentumLSTM in specific tasks. Computation time and memory cost of our models versus the baseline LSTM are provided in Appendix D. All of our results are averaged over 5 runs with different seeds. We include details on the models, datasets, training procedure, and hyperparameters used in our experiments in Appendix A. For MNIST and TIMIT experiments, we use the baseline codebase provided by [5]. For PTB experiments, we use the baseline codebase provided by [54].

## 3.1 Pixel-by-Pixel MNIST

In this task, we classify image samples of hand-written digits from the MNIST dataset [33] into one of the ten classes. Following the implementation of [32], we flatten the image of original size 28 × 28 pixels and feed it into the model as a sequence of length 784. In the unpermuted task (MNIST), the sequence of pixels is processed row-by-row. In the permuted task (PMNIST), a fixed permutation is selected at the beginning of the experiments and then applied to both training and test sequences. We summarize the results in Table 1. Our experiments show that *MomentumLSTM achieves better test accuracy than the baseline LSTM in both MNIST and PMNIST digit classification tasks* using different numbers of hidden units (i.e. $N = 128, 256$). Especially, the improvement is significant on the PMNIST task, which is designed to test the performance of RNNs in the context of long-term memory. Furthermore, we notice that *MomentumLSTM converges faster than LSTM* in all settings. Figure 3 (left two panels) corroborates this observation when using $N = 256$ hidden units.

## 3.2 TIMIT Speech Dataset

We study how MomentumLSTM performs on audio data with speech prediction experiments on the TIMIT speech dataset [16], which is a collection of real-world speech recordings. As first proposed by [62], the recordings are downsampled to 8kHz and then transformed into log-magnitudes via a short-time Fourier transform (STFT). The task accounts for predicting the next log-magnitude given the previous ones. We use the standard train/validation/test separation in [62, 34, 6], thereby having 3640 utterances for the training set with a validation set of size 192 and a test set of size 400.

The results for this TIMIT speech prediction are shown in Table 2. Results are reported on the test set using the model parameters that yield the best validation loss. Again, we see the advantage of MomentumLSTM over the baseline LSTM. In particular, MomentumLSTM yields much better prediction accuracy and faster convergence speed compared to LSTM. Figure 3 (right two panels) shows the convergence of MomentumLSTM vs. LSTM when using $N = 158$ hidden units.

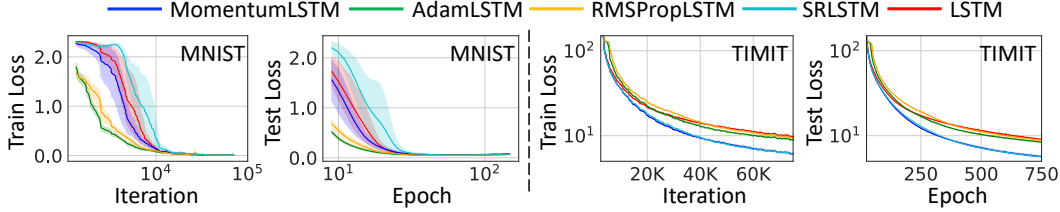

Figure 3: Train and test loss of MomentumLSTM (blue), AdamLSTM (green), RMSPropLSTM (orange), SRLSTM (cyan), and LSTM (red) for MNIST (left two panels) and TIMIT (right two panels) tasks. MomentumLSTM converges faster than LSTM in both tasks. For MNIST, AdamLSTM and RMSPropLSTM converge fastest. For TIMIT, MomentumLSTM and SRLSTM converge fastest.

Table 2: Test and validation MSEs at the end of the epoch with the lowest validation MSE for the TIMIT task. All of our proposed models outperform the baseline LSTM. Among models using $N = 158$ hidden units, SRLSTM performs the best.

| MODEL | N | # PARAMS | VAL. MSE | TEST MSE |
|---|---|---|---|---|
| LSTM | 84 | $\approx 83K$ | $14.87 \pm 0.15$ (15.42 [22, 34]) | $14.94 \pm 0.15$ (14.30 [22, 34]) |
| LSTM | 120 | $\approx 135K$ | $11.77 \pm 0.14$ (13.93 [22, 34]) | $11.83 \pm 0.12$ (12.95 [22, 34]) |
| LSTM | 158 | $\approx 200K$ | $9.33 \pm 0.14$ (13.66 [22, 34]) | $9.37 \pm 0.14$ (12.62 [22, 34]) |
| MOMENTUMLSTM | 84 | $\approx 83K$ | $\mathbf{10.90 \pm 0.19}$ | $\mathbf{10.98 \pm 0.18}$ |
| MOMENTUMLSTM | 120 | $\approx 135K$ | $\mathbf{8.00 \pm 0.30}$ | $\mathbf{8.04 \pm 0.30}$ |
| MOMENTUMLSTM | 158 | $\approx 200K$ | $\mathbf{5.86 \pm 0.14}$ | $\mathbf{5.87 \pm 0.15}$ |
| ADAMLSTM | 158 | $\approx 200K$ | $8.66 \pm 0.15$ | $8.69 \pm 0.14$ |
| RMSPROPLSTM | 158 | $\approx 200K$ | $9.13 \pm 0.33$ | $9.17 \pm 0.33$ |
| SRLSTM | 158 | $\approx 200K$ | $\mathbf{5.81 \pm 0.10}$ | $\mathbf{5.83 \pm 0.10}$ |

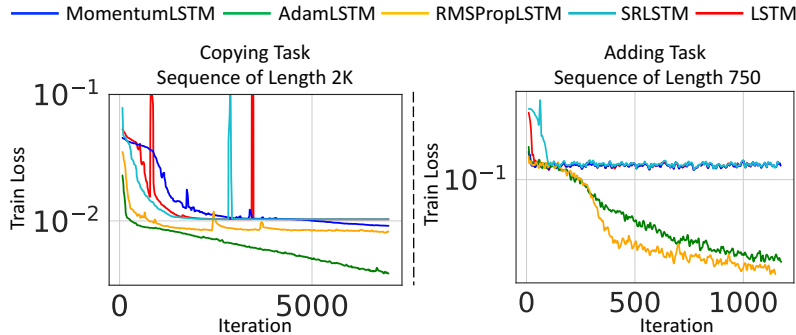

Figure 4: Train loss vs. iteration for (left) copying task with sequence length 2K and (right) adding task with sequence length 750. AdamLSTM and RMSPropLSTM converge faster and to better final losses than other models. MomentumLSTM and SRLSTM converge to similar losses as LSTM.

**Remark:** The TIMIT dataset is not open for public, so we do not have access to the preprocessed data from previous papers. We followed the data preprocessing in [62, 34, 6] to generate the preprocessed data for our experiments and did our best to reproduce the baseline results. In Table 2 and 5, we include both our reproduced results and the ones reported from previous works.

### 3.3 Copying and Adding Tasks

Two other important tasks for measuring the ability of a model to learn long-term dependency are the copying and adding tasks [24, 1]. In both copying and adding tasks, avoiding vanishing/exploding gradients becomes more relevant when the input sequence length increases. We compare the performance of MomentumLSTM over LSTM on these tasks. We also examine the performance of AdamLSTM, RMSPropLSTM, and SRLSTM on the same tasks. We define the copying and adding tasks in Appendix A.4 and summarize our results in Figure 4. In copying task for sequences of length 2K, MomentumLSTM obtains slightly better final training loss than the baseline LSTM (0.009 vs. 0.01). In adding task for sequence of length 750, both models achieve similar training loss of 0.162. However, AdamLSTM and RMSPropLSTM significantly outperform the baseline LSTM.

Table 3: Model test perplexity at the end of the epoch with the lowest validation perplexity for the Penn Treebank language modeling task (word level).

| MODEL | # PARAMS | VAL. PPL | TEST PPL |
|---|---|---|---|
| LSTM | $\approx 24M$ | $61.96 \pm 0.83$ | $59.71 \pm 0.99$ (58.80 [37]) |
| MOMENTUMLSTM | $\approx 24M$ | $\mathbf{60.71 \pm 0.24}$ | $\mathbf{58.62 \pm 0.22}$ |
| SRLSTM | $\approx 24M$ | $61.12 \pm 0.68$ | $58.83 \pm 0.62$ |

## 3.4 Word-Level Penn TreeBank

To study the advantage of MomentumLSTM over LSTM on text data, we perform language modeling on a preprocessed version of the PTB dataset [39], which has been a standard benchmark for evaluating language models. Unlike the baselines used in the (P)MNIST and TIMIT experiments which contain one LSTM cell, in this PTB experiment, we use a three-layer LSTM model, which contains three concatenated LSTM cells, as the baseline. The size of this model in terms of the number of parameters is also much larger than those in the (P)MNIST and TIMIT experiments. Table 3 shows the test and validation perplexity (PPL) using the model parameters that yield the best validation loss. Again, MomentumLSTM achieves better perplexities and converges faster than the baseline LSTM (see Figure 5).

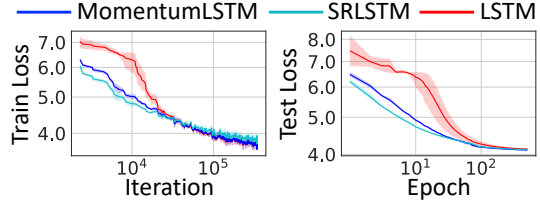

Figure 5: Train (left) and test loss (right) of MomentumLSTM (blue), SRLSTM (cyan), and LSTM (red) for the Penn Treebank language modeling tasks at word level.

## 3.5 NAG and Adam Principled Recurrent Neural Nets

We evaluate AdamLSTM, RMSPropLSTM and SRLSTM on all tasks. For (P)MNIST and TIMIT tasks, we summarize the test accuracy of the trained models in Tables 1 and 2 and provide the plots of train and test losses in Figure 3. We observe that though AdamLSTM and RMSPropLSTM work better than the MomentumLSTM at (P)MNIST task, they yield worse results at the TIMIT task. Interestingly, SRLSTM shows an opposite behavior - better than MomentunLSTM at TIMIT task but worse at (P)MNIST task. For the copying and adding tasks, Figure 4 shows that AdamLSTM and RMSPropLSTM converge faster and to better final training loss than other models in both tasks. Finally, for the PTB task, both MomentumLSTM and SRLSTM outperform the baseline LSTM (see Figure 5 and Table 3). However, in this task, AdamLSTM and RMSPropLSTM yields slightly worse performance than the baseline LSTM. In particular, test PPL for AdamLSTM and RMSPropLSTM are $61.11 \pm 0.31$, and $64.53 \pm 0.20$, respectively, which are higher than the test PPL for LSTM ($59.71 \pm 0.99$). We observe that there is no model that win in all tasks. This is somewhat expected, given the connection between our model and its analogy to optimization algorithm. An optimizer needs to be chosen for each particular task, and so is for our MomentumRNN. All of our models outperform the baseline LSTM.

## 4 Additional Results and Analysis

**Beyond LSTM.** Our interpretation of hidden state dynamics in RNNs as GD steps and the use of momentum to accelerate the convergence speed and improve the generalization of the model apply to many types of RNNs but not only LSTM. We show the applicability of our momentum-based design approach beyond LSTM by performing PMNIST and TIMIT experiments using the orthogonal RNN equipped with dynamic trivialization (DTRIV) [6]. DTRIV is currently among state-of-the-art models for PMNIST digit classification and TIMIT speech prediction tasks. Tables 4 and 5 consist of results for our method, namely MomentumDTRIV, in comparison with the baseline results. Again, MomentumDTRIV outperforms the baseline DTRIV by a margin in both PMNIST and TIMIT tasks while converging faster and overfitting less (see Figure 6). Results for AdamDTRIV, RMSPropDTRIV, and SRDTRIV on the PMNIST task are provided in Appendix C.

**Computational Time Comparison.** We study the computational efficiency of the proposed momentum-based models by comparing the time for our models to reach the same test accuracy for LSTM. When training on the PMNIST task using 256 hidden units, we observe that to reach 92.29% test accuracy for LSTM, LSTM needs 767 min while MomentumLSTM, AdamLSTM, RMSPropLSTM, and SRLSTM only need 551 min, **225min**, 416 min, and 348 min, respectively. More detailed results are provided in Appendix D.

Table 4: Best test accuracy on the PMNIST tasks (%) for MomentumDTRIV and DTRIV. We provide both our reproduced baseline results and those reported in [6]. MomentumDTRIV yields better results than the baseline DTRIV in all settings.

| N | # PARAMS | PMNIST (DTRIV) | PMNIST (MOMENTUMDTRIV) |
|---|---|---|---|
| 170 | $\approx 16K$ | $95.21 \pm 0.10$ (95.20 [6]) | $\mathbf{95.37 \pm 0.09}$ |
| 360 | $\approx 69K$ | $96.45 \pm 0.10$ (96.50 [6]) | $\mathbf{96.73 \pm 0.08}$ |
| 512 | $\approx 137K$ | $96.62 \pm 0.12$ (96.80 [6]) | $\mathbf{96.89 \pm 0.08}$ |

Table 5: Test and validation MSE of MomentumDTRIV vs. DTRIV at the epoch with the lowest validation MSE for the TIMIT task. MomentumDTRIV yields much better results than DTRIV.

| MODEL | N | # PARAMS | VAL. MSE | TEST MSE |
|---|---|---|---|---|
| DTRIV | 224 | $\approx 83K$ | $4.74 \pm 0.06$ (4.75 [6]) | $4.70 \pm 0.07$ (4.71 [6]) |
| DTRIV | 322 | $\approx 135K$ | $1.92 \pm 0.17$ (3.39 [6]) | $1.87 \pm 0.17$ (3.76 [6]) |
| MOMENTUMDTRIV | 224 | $\approx 83K$ | $\mathbf{3.10 \pm 0.09}$ | $\mathbf{3.06 \pm 0.09}$ |
| MOMENTUMDTRIV | 322 | $\approx 135K$ | $\mathbf{1.21 \pm 0.05}$ | $\mathbf{1.17 \pm 0.05}$ |

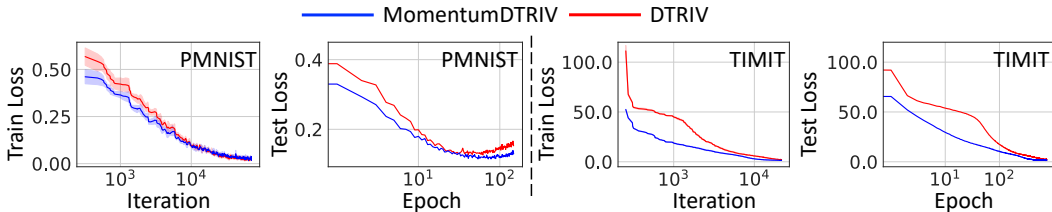

Figure 6: Train and test loss of MomentumDTRIV (blue) and DTRIV (red) for PMNIST (left two panels) and TIMIT (right two panels) tasks. MomentumDTRIV converges faster than DTRIV in both tasks. For PMNIST task, DTRIV suffers from overtting while MomentumDTRIV overfits less.

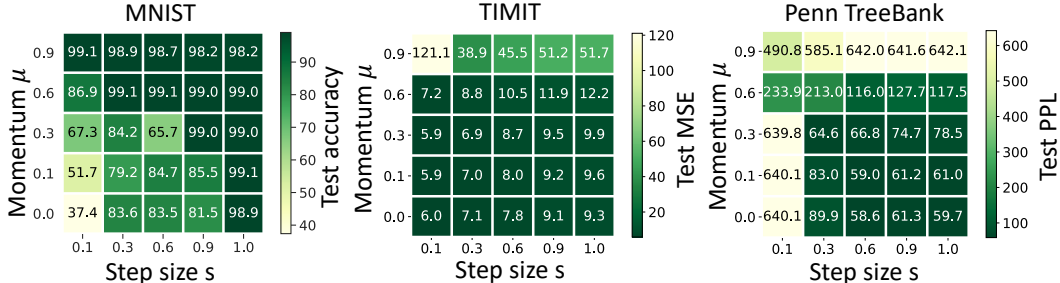

Figure 7: Ablation study of the effects of momentum and step size on MomentumLSTM's performance. We use $N = 256/158$ hidden units for MNIST/TIMIT task. Green denotes better results.

**Effects of Momentum and Step Size.** To better understand the effects of momentum and step size on the final performance of the trained MomentumLSTM models, we do an ablation study and include the results in Figure 7. The result in each cell is averaged over 5 runs.

## 5 Conclusion

In this paper, we propose a universal framework for integrating momentum into RNNs. The resulting MomentumRNN achieves significant acceleration in training and remarkably better performance on the benchmark sequential data prediction tasks over the RNN counterpart. From a theoretical viewpoint, it would be interesting to derive a theory to decipher why training MomentumRNN converges faster and generalizes better. From the neural architecture design perspective, it would be interesting to integrate momentum into the design of the standard convolutional and graph convolutional neural nets. Moreover, the current MomentumRNN requires calibration of the momentum and step size-related hyperparameters; developing an adaptive momentum for MomentumRNN is of interest.

## 6 Broader Impact and Ethical Considerations

Recurrent neural net (RNN) is among the most important classes of deep learning models. Improving training efficiency and generalization performance of RNNs not only advances image classification and language modeling but also benefits epidemiological models for pandemic disease prediction. RNNs have also been successfully used for the molecular generation [29]. Developing better RNNs that enable modeling of long term dependency, such as our Momentum RNN, has the potential to facilitate life science research. In order to fullfill that potential, more development is needed. For example, the current MomentumRNN requires calibration of the momentum and step size-related hyperparameters; developing an adaptive momentum for MomentumRNN is of great research interest. Finally, we claim that this paper does not have any ethical issue or leverage biases in data.

## 7 Acknowledgement

This material is based on research sponsored by the NSF grant DMS-1924935 and DMS-1952339, and the DOE grant DE-SC0021142. Other grants that support the work include the NSF grants CCF-1911094, IIS-1838177, and IIS-1730574; the ONR grants N00014-18-12571 and N00014-17-1-2551; the AFOSR grant FA9550-18-1-0478; the DARPA grant G001534-7500; and a Vannevar Bush Faculty Fellowship, ONR grant N00014-18-1-2047.

This material is also based upon work supported by the NSF under Grant# 2030859 to the Computing Research Association for the CIFellows Project, the NSF Graduate Research Fellowship Program, and the NSF IGERT Training Grant (DGE-1250104).

## Footnotes

*Please correspond to: wangbaonj@gmail.com or mn15@rice.edu

[2]In the vanishing gradient scenario, $\|\mathbf{U}\|_2$ is small; also it can be controlled by regularizing the loss function.

[3]In contrast to Adam, we do not normalize $\boldsymbol{p}_t$ and $\boldsymbol{m}_t$ since they can be absorbed in the weight matrices.

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
