[Supplementary Material]

**Appendix for "MomentumRNN: Integrating Momentum into Recurrent Neural Networks"**

# A  Experimental Details

In this section, we describe the datasets used in our experiments and provide details on the model implementation and training. MomentumLSTM, AdamLSTM, RMSPropLSTM, and SRLSTM, as well as MomentumDTRIV, AdamDTRIV, RMSPropDTRIV, and SRDTRIV share the same settings as their LSTM/DTRIV counterparts with the additional momentum $\mu$, step size $s$, scheduled restart $F$, and the coefficient $\beta$ used for computing running averages of the squared gradients. Thus, we only provide implementation and training details for the baseline LSTM and DTRIV for each task. Values for additional hyperparameters in our momentum-based models are found by grid search and reported in Table 7, 8, 9, and 10.

## A.1  Pixel-by-Pixel MNIST

MNIST dataset [33] consists of 60K training images and 10K test images from 10 classes of handwritten digits. Both training and test data are binary images of size $28 \times 28$. As mentioned in Section 3.1, we flatten and process the image as a sequence of the length of 784 pixel-by-pixel. In the unpermuted task (MNIST), the images are processed row-by-row, while in the permuted task (PMNIST), a fixed permutation is applied to both training and test images.

**LSTM.** The baseline LSTM models consist of one LSTM cell with 128 and 256 hidden units. Orthogonal initialization is used for input-to-hidden weights, while hidden-to-hidden weights are initialized to identity matrices. The forget gate bias is initialized to 1 while all other bias scalars are initialized to 0. We follow LSTM training in [34, 6] to train LSTM models for the MNIST and PMNIST tasks. Gradient norms are clipped to 1 during training, and the smoothing constant $\alpha$ for the RMSProp optimizer is set to 0.9. We provide other details on hyperparameters for the LSTM training on (P)MNIST in Table 6 (top).

**DTRIV.** We use the best DTRIV models for each (P)MNIST task reported in [6] with Cayley initialization [22]. The gradient norms are clipped to 1 during training. Other hyperparameter details are provided in Table 6 (bottom).

## A.2  TIMIT Speech Dataset

TIMIT speech dataset is a collection of real-world speech recordings [16] consisting of 3640 utterances for the training set, 192 utterances for the validation set, and 400 utterances for the test set. We follow the data preprocessing in [62, 6, 34, 22]. In particular, audio files in TIMIT are downsampled to 8kHz. A short-time Fourier transform (STFT) is then applied with a Hann window of 256 samples and a window hop of 128 samples (16 milliseconds) to yield sequences of 129 complex-valued Fourier amplitudes. The log-magnitude of these sequences is fed into the models as the input data. The task is to predict the next log-magnitude given the previous ones.

**LSTM.** The baseline LSTM models consist of one LSTM cell with 84, 120, and 158 hidden units. Similar to (P)MNIST experiments, orthogonal initialization is used for input-to-hidden weights, while hidden-to-hidden weights are initialized to identity matrices. However, the forget gate bias is initialized to -4 while all other bias scalars are initialized to 0. We follow LSTM training in [34, 6] to train LSTM models for the TIMIT tasks. We use the standard Adam optimizer in PyTorch [48] to train the models without using gradient clipping. We provide other details on hyperparameters for the LSTM training on TIMIT in Table 6 (top).

**DTRIV.** We use the best DTRIV models for each TIMIT task reported in [6] with Henaff initialization [23]. Other hyperparameter details are provided in Table 6 (bottom).

## A.3  Word-Level Penn TreeBank

The Penn TreeBank (PTB) dataset is among the most popular datasets for experimenting with language modeling. The dataset has 10,000 unique words and is preprocessed to not include capital letters, numbers, or punctuation [39].

**LSTM.** The baseline are three-layer LSTM models with 1150 hidden units at each layer and an embedding of size 400. We follow the LSTM implementation and training in [37]. We summarize some important details in Table 6 (top).

## A.4  Copying and Adding Tasks

We define the copying and adding tasks in Section 3.3 as follows.

**Copying task.** In the copying task, we consider a set $A$ of $N$ alphabet, e.g. $A = \{a_k\}_{k=1}^N$, and let `<start>` and `<blank>` be two symbols not contained in $A$. For a sequence of $K$ ordered characters sampled i.i.d. uniformly from $A$ and a spacing length L, the input sequence is the $K$ characters followed by $L$ `<blank>` characters, a `<start>` character, and then $K-1$ `<blank>` characters. The task is to output a sequence containing $K+L$ `<blank>` characters followed by the alphabet character sequence of length $K$. For example, let $A = \{1, 2, 3, 4\}$, $K = 5$, $L = 20$, `<start>` =:, and `<blank>` = $-$, an input sequence and its corresponding output sequence is given in Figure 8.

Figure 8: An example of input and output in the copying task.

**Adding task.** We follow the adding problem as proposed in [1], which is a variation of the similar problem in [24]. In particular, in this task, two sequences of length T are concurrently passed into an RNN. The first sequence consists of ordered digits sampled uniformly from a half-open interval $U[0, 1]$. The second sequence contains all zeros except for two entries that are marked by 1. The location of the first and second 1 is uniformly chosen within the interval $[1, T/2]$ and $[T/2, T]$, respectively. We label each pair of sequences by the sum of the two entries in the first sequence that are marked by 1's in the second sequence.

**LSTM.** The baseline LSTM models for the copying and adding tasks consist of one LSTM cell with 190 and 128 hidden units, respectively. Orthogonal initialization is used for input-to-hidden weights, while hidden-to-hidden weights are initialized to identity matrices. The forget gate bias is initialized to 1 while all other bias scalars are initialized to 0. We follow LSTM training in [34] and [35] to train LSTM models for the copying and adding tasks, respectively. We provide details on hyperparameters for the LSTM training on the copying task in Table 6 (top).

### A.5 Momentum Cells can Avoid Vanishing Gradient Issue

To confirm that MomentumRNN can alleviate vanishing gradients, we train a MomentumDTRIV and its corresponding baseline DTRIV for the PMNIST classification task. We plot $\|\partial \mathcal{L}/\partial \boldsymbol{h}_t\|_2$ for each time step $t$ at each training iteration, as shown in Figure 2. Both MomentumDTRIV and DTRIV models used in this experiment contains one cell of 170 hidden units. The model implementation and training details are similar to those in Section A.1 above. Note that DTRIV is also an RNN with additional orthogonality constraint.

## B  Backpropagation Through Time – A Review

In this section, we give a short review of the backpropagation through time, which is a major algorithm for training RNNs. We consider the standard recurrent cell (1), and for any given training sample $(\boldsymbol{x}, \boldsymbol{y})$ with $\boldsymbol{x} = (\boldsymbol{x}_1, \cdots, \boldsymbol{x}_T)$ being an input sequence of length $T$ and $\boldsymbol{y} = (y_1, \cdots, y_T)$ being the sequence of labels [4]. Let $\mathcal{L}_t$ be the loss at the time step $t$ and the total loss on the whole sequence is

$$\mathcal{L} = \sum_{t=1}^{T} \mathcal{L}_t. \tag{16}$$

For any $1 \leq t \leq T$, we can compute the gradient of the loss $\mathcal{L}_t$ with respect to the parameter $\mathbf{U}$ as

$$\frac{\partial \mathcal{L}_t}{\partial \mathbf{U}} = \sum_{k=1}^{t} \frac{\partial \boldsymbol{h}_k}{\partial \mathbf{U}} \cdot \frac{\partial \mathcal{L}_t}{\partial \boldsymbol{h}_t} \cdot \frac{\partial \boldsymbol{h}_t}{\partial \boldsymbol{h}_k} = \sum_{k=1}^{t} \frac{\partial \boldsymbol{h}_k}{\partial \mathbf{U}} \cdot \frac{\partial \mathcal{L}_t}{\partial \boldsymbol{h}_t} \cdot \prod_{k=1}^{t-1} \frac{\partial \boldsymbol{h}_{k+1}}{\partial \boldsymbol{h}_k}, \tag{17}$$

where $\frac{\partial \boldsymbol{h}_{k+1}}{\partial \boldsymbol{h}_k} = \mathbf{D}_k \mathbf{U}^{\mathrm{T}}$ with $\mathbf{D}_k = \mathrm{diag}(\sigma'(\mathbf{U}\boldsymbol{h}_k + \mathbf{W}\boldsymbol{x}_{k+1} + \boldsymbol{b}))$. Similarly, we can compute $\partial \mathcal{L}_t/\partial \mathbf{W}$ and $\partial \mathcal{L}_t/\partial \boldsymbol{b}$.

Table 6: Hyperparameters for the Baseline LSTM and DTRIV Training.

## LSTM

| Dataset | Optimizer | Learning Rate | Batch Size | #Epochs |
|---|---|---|---|---|
| MNIST | RMSProp | 0.001 | 128 | 150 |
| PMNIST | RMSProp | 0.001 | 128 | 150 |
| TIMIT | Adam | 0.0001 | 32 | 700 |
| PTB | SGD | 30 (initial learning rate) | 20 | 500 |
| Copying | RMSprop | 0.0002 | 128 | 7000 |
| Adding | Adam | 0.0002 | 50 | 1200 |

## DTRIV

| Dataset | Size | DTRIV Opt. Step (K) | Optimizer | Learning Rate | Orthogonal Optimizer | Orthogonal Learning Rate | Batch Size | #Epochs |
|---|---|---|---|---|---|---|---|---|
| MNIST | 170 | 1 | | 0.001 | | 0.0001 | 128 | 150 |
| MNIST | 360 | ∞ | RMSProp | 0.0005 | RMSProp | 0.0001 | 128 | 150 |
| MNIST | 512 | 100 | | 0.0005 | | 0.0001 | 128 | 150 |
| PMNIST | 170 | 1 | | 0.0007 | | 0.0002 | 128 | 150 |
| PMNIST | 360 | ∞ | RMSProp | 0.0007 | RMSProp | 0.00005 | 128 | 150 |
| PMNIST | 512 | ∞ | | 0.0003 | | 0.00007 | 128 | 150 |
| TIMIT | 224 | ∞ | Adam | 0.001 | RMSProp | 0.0002 | 128 | 700 |
| TIMIT | 322 | ∞ | | 0.001 | | 0.0002 | 128 | 700 |

Table 7: Hyperparameters for MomentumLSTM and MomentumDTRIV Training

## MomentumLSTM

| Dataset | Momentum $\mu$ | Step Size $s$ | Optimizer | Learning Rate | Batch Size | #Epochs |
|---|---|---|---|---|---|---|
| MNIST | 0.6 | 0.6 | RMSProp | 0.001 | 128 | 150 |
| PMNIST | 0.6 | 1.0 | RMSProp | 0.001 | 128 | 150 |
| TIMIT | 0.3 | 0.1 | Adam | 0.0001 | 32 | 700 |
| PTB | 0.0 | 0.6 | SGD | 30 (initial learning rate) | 20 | 500 |
| Copying (sequence length 1K) | 0.6 | 0.9 | RMSprop | 0.0002 | 128 | 7000 |
| Copying (sequence length 2K) | 0.9 | 2.0 | RMSprop | 0.0002 | 128 | 7000 |
| Adding | 0.9 | 2.0 | Adam | 0.0002 | 50 | 1200 |

## MomentumDTRIV

| Dataset | Size | DTRIV Opt. Step (K) | Momentum $\mu$ | Step Size $s$ | Optimizer | Learning Rate | Orthogonal Optimizer | Orthogonal Learning Rate | Batch Size | #Epochs |
|---|---|---|---|---|---|---|---|---|---|---|
| PMNIST | 170 | 1 | 0.6 | 0.9 | | 0.0007 | | 0.0002 | 128 | 150 |
| PMNIST | 360 | ∞ | 0.3 | 0.3 | RMSProp | 0.0007 | RMSProp | 0.00005 | 128 | 150 |
| PMNIST | 512 | ∞ | 0.3 | 0.3 | | 0.0003 | | 0.00007 | 128 | 150 |
| TIMIT | 224 | ∞ | 0.3 | 0.1 | Adam | 0.001 | RMSProp | 0.0002 | 128 | 700 |
| TIMIT | 322 | ∞ | 0.3 | 0.1 | | 0.001 | | 0.0002 | 128 | 700 |

Table 8: Hyperparameters for AdamLSTM and AdamDTRIV Training

## AdamLSTM

| Dataset | Optimizer | Momentum $\mu$ | Step Size $s$ | $\beta$ | Learning Rate | Batch Size | #Epochs |
|---|---|---|---|---|---|---|---|
| MNIST | RMSProp | 0.6 | 0.6 | 0.1 | 0.001 | 128 | 150 |
| PMNIST | RMSProp | 0.6 | 1.0 | 0.01 | 0.001 | 128 | 150 |
| TIMIT | Adam | 0.3 | 0.1 | 0.999 | 0.0001 | 32 | 700 |
| Copying (sequence length 1K) | RMSprop | 0.6 | 2.0 | 0.999 | 0.0002 | 128 | 7000 |
| Copying (sequence length 2K) | RMSprop | 0.6 | 2.0 | 0.999 | 0.0002 | 128 | 7000 |
| Adding | Adam | 0.6 | 2.0 | 0.999 | 0.0002 | 50 | 1200 |

## AdamDTRIV

| Dataset | Size | DTRIV Opt. Step (K) | Momentum $\mu$ | Step Size $s$ | $\beta$ | Optimizer | Learning Rate | Orthogonal Optimizer | Orthogonal Learning Rate | Batch Size | #Epochs |
|---|---|---|---|---|---|---|---|---|---|---|---|
| PMNIST | 512 | ∞ | 0.3 | 0.3 | 0.8 | RMSProp | 0.0003 | RMSProp | 0.00007 | 128 | 150 |

Table 9: Hyperparameters for RMSPropLSTM and RMSPropDTRIV Training

RMSPropLSTM

| Dataset | Optimizer | Momentum $\mu$ | Step Size $s$ | $\beta$ | Learning Rate | Batch Size | #Epochs |
|---|---|---|---|---|---|---|---|
| MNIST | RMSProp | 0.0 | 0.6 | 0.9 (size $N = 256$) 0.99 (size $N = 128$) | 0.001 | 128 | 150 |
| PMNIST | RMSProp | 0.0 | 1.0 | 0.01 | 0.001 | 128 | 150 |
| TIMIT | Adam | 0.0 | 0.1 | 0.999 | 0.0001 | 32 | 700 |
| Copying (sequence length 1K) | RMSprop | 0.0 | 2.0 | 0.999 | 0.0002 | 128 | 7000 |
| Copying (sequence length 2K) | RMSprop | 0.0 | 2.0 | 0.999 | 0.0002 | 128 | 7000 |
| Adding | Adam | 0.0 | 2.0 | 0.999 | 0.0002 | 50 | 1200 |

RMSPropDTRIV

| Dataset | Size | DTRIV Opt. Step (K) | Momentum $\mu$ | Step Size $s$ | $\beta$ | Optimizer | Learning Rate | Orthogonal Optimizer | Orthogonal Learning Rate | Batch Size | #Epochs |
|---|---|---|---|---|---|---|---|---|---|---|---|
| PMNIST | 512 | $\infty$ | 0.0 | 0.3 | 0.9 | RMSProp | 0.0003 | RMSProp | 0.00007 | 128 | 150 |

Table 10: Hyperparameters for SRLSTM and SRDTRIV Training

SRLSTM

| Dataset | Optimizer | Scheduled Restart (F) | Step Size $s$ | Learning Rate | Batch Size | #Epochs |
|---|---|---|---|---|---|---|
| MNIST | RMSProp | 2 | 1.0 | 0.001 | 128 | 150 |
| PMNIST | RMSProp | 40 (size $N = 256$) 6 (size $N = 128$) | 0.9 (size $N = 256$) 0.01 (size $N = 128$) | 0.001 | 128 | 150 |
| TIMIT | Adam | 2 | 0.1 | 0.0001 | 32 | 700 |
| PTB | SGD | 2 | 0.6 | 30 (initial learning rate) | 20 | 500 |
| Copying (sequence length 1K) | RMSProp | 100 | 0.9 | 0.0002 | 128 | 7000 |
| Copying (sequence length 2K) | RMSProp | 100 | 0.9 | 0.0002 | 128 | 7000 |
| Adding | Adam | 100 | 0.9 | 0.0002 | 50 | 1200 |

SRDTRIV

| Dataset | Size | DTRIV Opt. Step (K) | Scheduled Restart (F) | Step Size $s$ | Optimizer | Learning Rate | Orthogonal Optimizer | Orthogonal Learning Rate | Batch Size | #Epochs |
|---|---|---|---|---|---|---|---|---|---|---|
| PMNIST | 512 | $\infty$ | 2 | 0.3 | RMSProp | 0.0003 | RMSProp | 0.00007 | 128 | 150 |

# C  More Experimental Results

We conduct more comprehensive experiments for the Adam principled and NAG principled RNNs. In particular, we perform (P)MNIST and TIMIT experiments using the AdamLSTM, RMSPropLSTM, and SRLSTM of 128 and 120 hidden units, respectively. For (P)MNIST task, RMSPropLSTM achieves the best test accuracy and converges the fastest. For the TIMIT task, MomentumLSTM and SRLSTM outperform the other models while converging faster. We summarize our results in Table 11 and 12, as well as in Figure 9. Note that in the main text, we conduct the same experiments using the same models but with different numbers of hidden units (i.e. 256 hidden units for the (P)MNIST task and 158 hidden units for the TIMIT task).

Furthermore, we provide additional results on copying task for sequences of length 1K in comparison with those for sequences of length 2K as in the main text. In addition to training losses, we also include test losses in in Figure 10.

Finally, we apply our Adam and NAG principled designing methods on a DTRIV, an orthogonal RNN [6], for the PMNIST classification task. We observe that AdamDTRIV, RMSPropDTRIV, and SRDTRIV outperform the baseline DTRIV while converging faster. SRDTRIV also outperforms MomentumDTRIV. We summarize our results in Table 13 and Figure 11. Hyperparameter values for this experiment can be found in Table 8, 9, and 10 (bottom).

Table 11: Best test accuracy at the MNIST and PMNIST tasks (%). We use the baseline results reported in [22], [62], [60]. All of our proposed models outperform the baseline LSTM. Among the models using $N = 128$ hidden units, RMSPropLSTM yields the best results in both tasks.

| MODEL | N | # PARAMS | MNIST | PMNIST |
|---|---|---|---|---|
| LSTM | 128 | $\approx 68K$ | 98.70[22],97.30 [60] | 92.00 [22],92.62 [60] |
| MOMENTUMLSTM | 128 | $\approx 68K$ | **99.04 ± 0.04** | **93.40 ± 0.25** |
| ADAMLSTM | 128 | $\approx 68K$ | 98.98 ± 0.08 | 93.75 ± 0.25 |
| RMSPROPLSTM | 128 | $\approx 68K$ | **99.09 ± 0.05** | **94.32 ± 0.43** |
| SRLSTM | 128 | $\approx 68K$ | 98.89 ± 0.08 | 93.65 ± 0.56 |

Table 12: Test and validation MSEs at the end of the epoch with the lowest validation MSE for the TIMIT task. All of our proposed models outperform the baseline LSTM. Among models using $N = 120$ hidden units, MomentumLSTM performs the best.

| MODEL | N | # PARAMS | VAL. MSE | TEST MSE |
|---|---|---|---|---|
| LSTM | 120 | $\approx 135K$ | 11.77 ± 0.14 (13.93 [22, 34]) | 11.83 ± 0.12 (12.95 [22, 34]) |
| MOMENTUMLSTM | 120 | $\approx 135K$ | **8.00 ± 0.30** | **8.04 ± 0.30** |
| ADAMLSTM | 120 | $\approx 135K$ | 10.91 ± 0.08 | 10.96 ± 0.08 |
| RMSPROPLSTM | 120 | $\approx 135K$ | 11.83 ± 0.20 | 11.90 ± 0.19 |
| SRLSTM | 120 | $\approx 135K$ | 8.15 ± 0.26 | 8.21 ± 0.26 |

Figure 9: Train and test loss of MomentumLSTM (blue), AdamLSTM (green), RMSPropLSTM (orange), SRLSTM (cyan), and LSTM (red) using $N = 128$ hidden units for MNIST (left two panels) and using $N = 120$ hidden units for TIMIT (right two panels) tasks. MomentumLSTM converges faster than LSTM in both tasks. RMSPropLSTM and MomentumLSTM/SRLSTM converge the fastest for MNIST and TIMIT tasks, respectively.

Figure 10: Train test loss vs. iteration for copying task with sequence length 1K (left) and 2K (right). AdamLSTM and RMSPropLSTM converge faster and to better final losses than other models. MomentumLSTM and SRLSTM converge to similar losses as LSTM.

Table 13: Best test accuracy on the PMNIST tasks (%) for MomentumDTRIV and the baseline DTRIV, as well as for AdamDTRIV, RMSPropDTRIV, and SRDTRIV. We provide both our reproduced baseline results and those reported in [6]. All of our momentum-based models outperform the baseline DTRIV. When using $N = 512$ hidden units, SRDTRIV yields the best result.

| MODEL | N | # PARAMS | PMNIST |
|---|---|---|---|
| DTRIV | 170 | $\approx 16K$ | $95.21 \pm 0.10$ (95.20 [6]) |
| DTRIV | 360 | $\approx 69K$ | $96.45 \pm 0.10$ (96.50 [6]) |
| DTRIV | 512 | $\approx 137K$ | $96.62 \pm 0.12$ (96.80 [6]) |
| MOMENTUMDTRIV | 170 | $\approx 16K$ | $\mathbf{95.37 \pm 0.09}$ |
| MOMENTUMDTRIV | 360 | $\approx 69K$ | $\mathbf{96.73 \pm 0.08}$ |
| MOMENTUMDTRIV | 512 | $\approx 137K$ | $\mathbf{96.89 \pm 0.08}$ |
| ADAMDTRIV | 512 | $\approx 137K$ | $\mathbf{96.77 \pm 0.21}$ |
| RMSPROPDTRIV | 512 | $\approx 137K$ | $\mathbf{96.75 \pm 0.12}$ |
| SRDTRIV | 512 | $\approx 137K$ | $\mathbf{97.02 \pm 0.09}$ |

Figure 11: Train and test loss of MomentumDTRIV (blue), AdamDTRIV (green), RMSPropDTRIV (orange), SRDTRIV (cyan), and DTRIV (red) for PMNIST task. Our momentum-based models converge faster than the baseline DTRIV.

# D  Computational Time and Memory Cost: RNN vs. MomentumRNN

We provide the computation time and memory cost per sample at training and evaluation of MomentumLSTM, AdamLSTM, RMSPropLSTM, and SRLSTM in comparison with LSTM for PMNIST classification task using 256 hidden units in Table 14 and 15, respectively.

Table 14: Computation time per sample at training and evaluation for PMNIST classification task using models with 256 hidden units.

| MODEL | TRAINING TIME ($\mu s$/SAMPLE) | EVALUATION TIME ($\mu s$/SAMPLE) |
|---|---|---|
| LSTM | 6.18 | 2.52 |
| MOMENTUMLSTM | 7.43 | 3.16 |
| ADAMLSTM | 10.34 | 4.07 |
| RMSPROPLSTM | 9.94 | 3.96 |
| SRLSTM | 8.34 | 3.16 |

Table 15: Memory cost per sample at training and evaluation for PMNIST classification task using models with 256 hidden units.

| MODEL | TRAINING MEMORY (MB/SAMPLE) | EVALUATION MEMORY (MB/SAMPLE) |
|---|---|---|
| LSTM | 15.93 | 7.51 |
| MOMENTUMLSTM | 15.95 | 7.51 |
| ADAMLSTM | 25.13 | 7.52 |
| RMSPROPLSTM | 25.13 | 7.52 |
| SRLSTM | 15.95 | 7.51 |

Table 16: Total computation time to reach the same 92.29% test accuracy of LSTM (see Tab. 1) for PMNIST classification task using models with 256 hidden units.

| MODEL | TIME ($min$) |
|---|---|
| LSTM | 767 |
| MOMENTUMLSTM | 551 |
| ADAMLSTM | **225** |
| RMSPROPLSTM | 416 |
| SRLSTM | 348 |

## E  Additional Information about the Figures in the Main Text

In Figure 3, the MNIST plots are for models with 256 hidden units, and the TIMIT plots are for models with 158 hidden units.

In Figure 6, the PMNIST plots are for models with 512 hidden units, and the TIMIT plots are for models with 322 hidden units.

## F  MomentumLSTM Cell Implementation in Pytorch

```python
import torch
import torch.nn as nn
from torch.nn import functional as F

class MomentumLSTMCell(nn.Module):

    """
    An implementation of MomentumLSTM Cell

    Args:
        input_size: The number of expected features in the input 'x'
        hidden_size: The number of features in the hidden state 'h'
        mu: momentum coefficient in MomentumLSTM Cell
        s: step size in MomentumLSTM Cell
        bias: If ''False'', then the layer does not use bias weights '
            b_ih' and 'b_hh'. Default: ''True''

    Inputs: input, hidden0=(h_0, c_0), v0
        - input of shape '(batch, input_size)': tensor containing input
            features
        - h_0 of shape '(batch, hidden_size)': tensor containing the
            initial hidden state for each element in the batch.
        - c_0 of shape '(batch, hidden_size)': tensor containing the
            initial cell state for each element in the batch.
        - v0 of shape '(batch, hidden_size)': tensor containing the
            initial momentum state for each element in the batch

    Outputs: h1, (h_1, c_1), v1
        - h_1 of shape '(batch, hidden_size)': tensor containing the next
            hidden state for each element in the batch
        - c_1 of shape '(batch, hidden_size)': tensor containing the next
            cell state for each element in the batch
        - v_1 of shape '(batch, hidden_size)': tensor containing the next
            momentum state for each element in the batch
    """

    def __init__(self, input_size, hidden_size, mu, s, bias=True):
        super(MomentumLSTMCell, self).__init__()
        self.input_size = input_size
        self.hidden_size = hidden_size
        self.bias = bias
        self.x2h = nn.Linear(input_size, 4 * hidden_size, bias=bias)
        self.h2h = nn.Linear(hidden_size, 4 * hidden_size, bias=bias)
```

```python
        # for momentumnet
        self.mu = mu
        self.s = s

        self.reset_parameters(hidden_size)

    def reset_parameters(self, hidden_size):
        nn.init.orthogonal_(self.x2h.weight)
        nn.init.eye_(self.h2h.weight)
        nn.init.zeros_(self.x2h.bias)
        self.x2h.bias.data[hidden_size:(2 * hidden_size)].fill_(1.0)
        nn.init.zeros_(self.h2h.bias)
        self.h2h.bias.data[hidden_size:(2 * hidden_size)].fill_(1.0)

    def forward(self, x, hidden, v):

        hx, cx = hidden

        x = x.view(-1, x.size(1))
        v = v.view(-1, v.size(1))

        vy = self.mu * v + self.s * self.x2h(x)

        gates = vy + self.h2h(hx)

        gates = gates.squeeze()

        ingate, forgetgate, cellgate, outgate = gates.chunk(4, 1)

        ingate = F.sigmoid(ingate)
        forgetgate = F.sigmoid(forgetgate)
        cellgate = F.tanh(cellgate)
        outgate = F.sigmoid(outgate)

        cy = torch.mul(cx, forgetgate) + torch.mul(ingate, cellgate)

        hy = torch.mul(outgate, F.tanh(cy))

        return hy, (hy, cy), vy
```

## Footnotes

[4]Without loss of generality, we consider the sequence to sequence modeling.