[Reviews · NeurIPS 2020]

Review 1

Summary and Contributions: This paper proposes a gradient descent view of RNN dynamics, and shows how to incorporate momentum for accelerating gradient dynamics. The proposed method can address the vanishing gradient problem. ============================= After Author Response: ============================= After reading the author response and discussing with the other reviewers, I am keeping my score. The main issue I have with this paper is that it makes speculative claims that are not adequately supported, but tries to pass them off as conclusive. Some examples include: (*) Phrases such as - "we establish a connection between the hidden state dynamics in an RNN and gradient descent (GD)" (Abstract) - "MomentumRNN is principled with theoretical guarantees provided by the momentum-accelerated dynamical system for optimization and sampling" (Contributions) - and several more throughout the paper are examples of misleading language in my opinion. (*) I mentioned concerns with the related work and background in the original review, which were not addressed in the author response. (*) The paper frequently mentions demonstrating that "based on our experiments, the acceleration from momentum is preserved". This was *not* demonstrated -- empirically faster convergence is not equatable to acceleration in an optimization sense. It is better to use precise language to separate what you have shown (an architecture with faster convergence) and what you are hypothesizing (it is related to momentum acceleration). As presented, I am not convinced of the latter connection. I think it's fine to say that your method is *inspired* by momentum, but in my opinion the paper implies a much stronger connection that is not substantiated by the theoretical and empirical results. I currently remain unconvinced that the proposed method's improvements are related to momentum at all. There are plenty of simpler explanations, as also offered by Reviewers 3 and 4, which should at least be discussed and ideally ablated. Ultimately, I see this paper as posing an interesting possible connection, but one that is currently speculative and not ready for publication. Aside from the overall writing, I have a few more detailed suggestions for improving the paper. (*) Although the paper positions itself in the space of "addressing the vanishing gradient issue in RNNs", this area is notably absent from the related work. For example, there are numerous such papers which operate on similar principled foundations, are much simpler, and perform better than the complicated models such as AdamLSTM proposed here. See e.g. [1, 2, 3] on PMNIST. These models are not difficult to implement, and it would be more convincing to use a more SOTA model instead of, or in addition to, the dtriv model used here. Other reviewers have also provided other connections and references. [1] Tallec, Olliver. Can RNNs Warp Time? [2] Gu et al. Improving the Gating Mechanism of RNNs [3] Voelker et al. Legendre Memory Units (*) Because I am not convinced that the benefits of the MomentumRNN cell come from momentum per se, the other cells like RMSPropLSTM/AdamLSTM seem very convoluted to me. The experimental results for them are not very convincing: the cell is so complicated that there are many uncontrolled architecture changes, which would benefit from an ablation study. (*) The authors claim that they "theoretically prove that MomentumRNNs alleviate the vanishing gradient issue" (Abstract), but there is no formal statement, much less a proof. As said in my initial review, the theory section merely concludes that "a \mu exists" without justification. Since \mu is the important momentum hyperparameter -- the main addition that this work proposes -- there is an opportunity to flesh out the theory and connect it to experiments. The fact that 2 out of 3 of the datasets are best when \mu is set to 0 is suspicious, and deserves further explanation. Regardless of whether this paper is accepted, I sincerely hope this feedback is useful to the authors and helps them refine the paper.

Strengths: - The explanation of the method is clear - The experimental results, including table and figures, are easy to understand - The method shows decent performance on the benchmarks, especially on the TIMIT task

Weaknesses: I have substantial concerns with the stated motivation and claims of the paper. Overall, the motivation and pitch of the paper imply a deeper connection to optimization that seems misguided at best, and the analysis, experiments, and baselines could be improved. - The authors main source of justification of the method is the claim that it is principled based on analogies to optimization methods. However, the analogies do not seem appropriate in this context in many ways, from the setting to the implementation details. As just one example, the additional linear and non-linear transformations (multiplication by U and a $\sigma$ nonlinearity, as in line 104) applied after the "momentum step" drastically alter the dynamics compared to momentum in optimization. The motivation, and even the name of the proposed method, feel misleading, supported only by a tenuous connection without deeper analysis. - The theory of the method is quite weak. The theoretical (Section 2.3) and empirical (Figure 2) analysis is with respect to a vanilla RNN, which is a poor baseline and as the authors note there are countless other works that address vanishing gradients. The analysis concludes that "there is an appropriate choice of \mu that can alleviate vanishing gradients", but this seems unsubstantiated; given the (implicit) constraint that $\mu$ is between 0 and 1, and the fact that $\Sigma_k$ can vary per timestep, it is not obvious that an appropriate choice of $\mu$ exists. - Additionally, the datasets used are quite toy, and lack any of the numerous baselines in the RNN family of methods besides the LSTM. For example, there are many similar RNN extensions that outperform this method on the MNIST/PMNIST benchmarks.

Correctness: The methods and experimental protocol seem correct. However, the claimed connection to optimization methods are not supported.

Clarity: The paper is overall well-written and easy to understand

Relation to Prior Work: The paper brings up many lines of related work. However, some of them seem less relevant (e.g. Langevin/Hamiltonian Monte Carlo and other optimization theory), while many of the more directly relevant works on improving long-term dependencies in RNNs are not discussed.

Reproducibility: Yes

Additional Feedback: It seems clear to me that the improvements are not from a magical connection to momentum in optimization, but simply from the addition of linear combination of updates $v_t = \mu v_{t-1} + s W x_t$, which add additive ("residual") connections to the state $v_t$ that allow gradients to backpropagate more easily. This is exactly analogous to the motivation of the gated update $c_t = f c_{t-1} + i h_t$ of the cell state of LSTMs, where $\mu, s$ take the role of the forget and input gates $f, i$. In fact, the main momentum cell (equation (8) or (10)) looks very similar to a LSTM with slight rewiring between the cell/hidden state, where $v_t$ plays the role of the cell state; the improvements of MomentumLSTM can be explained by its usage of multiple layers of linear "cell" states. These sorts of gating techniques and variants (e.g. a plethora of linear dynamics incorporated into RNNs) and their benefits for vanishing gradients has been well-studied, but the paper lacks a discussion and basic comparisons against such closely related and well-known methods. I believe such connections are obfuscated by the analogies to optimization. Simply incorporating a standard linear recurrence $v_t = \mu v_{t-1} + s W x_t$ does not beget a connection to momentum or other optimization principles.


Review 2

Summary and Contributions: The paper present a novel framework termed MomentumRNN. It is the incorporation of momentum into the connection of hidden states dynamics and gradient descent. The new model is presented in various forms and benchmarked on 3 different datasets, where it shows faster training, more robustness, and higher performance.

Strengths: - The paper does a thorough analysis on how to integrate various forms of momentum into different RNN types, and provides a simple yet powerful framework to do this. - The method and the mathematical basis to motivate these adaptions are provided. - The results show an increase in learning speed, more robustness against vanishing/exploding gradients, and improved final test accuracy. - Code examples allow the fast implementation of the framework into existing models and benchmarks. - Because of the widespread use of RNNs for all kinds of tasks make an optimization like MomentumRNN quite relevant.

Weaknesses: The new method requires additional parameters and fine-tuning of hyperparameters. Why this framework provides the aforementioned improvements is not clear. (Both of these points are mentioned in the conclusions by the authors as well)

Correctness: The claims seem correct, the empirical methodology in how the experiments were done and described are sound. An ablation study was also done for all 3 benchmarks.

Clarity: The clarity of the paper and its structure are good.

Relation to Prior Work: Previous contributions including s.o.t.a. approaches are correctly mentioned, described, and the differences to this new framework highlighted.

Reproducibility: Yes

Additional Feedback: An interesting metric would be to compare the computation (e.g. a rough FLOPs approximation) needed with this new method (that requires additional steps & parameters) to reach the same level of accuracy compared to a model without momentum that has to train for longer. Update: I changed the final score.


Review 3

Summary and Contributions: This paper makes an interesting connection between gradient updates in gradient descent and hidden state integration in RNNs. By making this connection, authors extend the idea of momentum in optimization to improve gradient flow in RNNs. This idea of momentum can be integrated into any RNN architecture and the authors show performance improvements in LSTMs and DTRIV by integrating momentum.

Strengths: 1. The paper makes a very novel connection between two ideas which is very thought-provoking! 2. The experiments prove that the proposed momentumRNN improves performance.

Weaknesses: 1. The authors could do a better job in motivating the model than simply saying this is a more principled way. I will give other motivations in the additional comments section.

Correctness: I do not see any incorrect claims. The experiments could be more controlled. Please refer to additional comments for suggestions.

Clarity: The paper is well written and I enjoyed reading this paper.

Relation to Prior Work: I appreciate an extensive coverage of 58 references. However, it would be nice if the authors also cover the line of work which discusses the issues with the saturations due to the gating mechanism in LSTMs (For example, JANET (Westhuizen and Lasenby 2018), NRU (Chandar et al 2019)).

Reproducibility: Yes

Additional Feedback: Important comments: 1. One way to motivate this work is as follows: In an LSTM, the cell state is trying to do additive integration of input so that gradients do not vanish. However, the sigmoid and tanh gates used in LSTM makes gradients vanish. Momentum RNN is trying to do additive integration of the input without any additional gates and hence has a much better gradient flow. Using MomentumRNN, or RMSPromRNN, or ADAMRNN is just equivalent to trying different integration schemes. And their performance might vary depending on the nature of the data. 2. While momentum is similar to what LSTM is trying to do with cell state integration, I am not sure why RMSPropLSTM should work. Because RMSProp is not a momentum method, it is an adaptive gradient method which adapts the learning rate automatically. Can you explain your intuitions why RMSPropLSTM should work? 3. Is there a reason you used different optimizers for different tasks? I am not sure if comparing your method to LSTM+SGD is fair since your model has some momentum integrated into it while LSTM+SGD does not. Using Adam for all your models, all your tasks might be a more controlled setup. I am inclined to accept the paper. So can you please give me these results in the rebuttal so that I can make up my mind? Other comments: 1. Please change the first 2 lines of the abstract. This work is not to overcome the expensive search. You can just say you find this novel connection. 2. Figure 2 is not meaningful without seeing the corresponding loss plots. The gradients could go to zero even when the model has solved the problem. 3. Line 123: I think MomentumLSTM should be explained in the main text since it gives an idea of how to integrate momentum to a non-vanilla RNN. 4. Equation 12 - shouldn’t the first + be - ? 5. Why did you use only the MomentumRNN in language modelling task? Is it because other models did not do well? I would like to see the performance of other models for reference. 6. Figure 6 is troubling. Based on the ablation, it looks like often momentum=0 is the best thing to do? Please elaborate more on Figure 6. You need to give me a strong justification for figure 6 to convince me to accept the paper. 7. Line 456 - why is forget gate initialized to -4? I have never seen this in any other work. Do you have a reference for others doing this initialization? Minor comments: 1. Line 45 - fix grammar “We then proposed to ..” 2. Line 115 - mention that mu and s are the hyperparameters. 3. Line 135 - fix “dominates” After Rebuttal: I am happy with the answers.


Review 4

Summary and Contributions: This paper introduces a new cell design for recurrent neural nets, which is inspired by the momentum in stochastic gradient decent-based optimization. The proposed method, called Momentum RNN, is able to alleviate the gradient vanishing problem and can be universally applied to a wide range of RNN structures. Momentum RNN is evaluated through several sequence modeling tasks.

Strengths: - The topic is of interest to majority of machine learning community. Since gradient vanishing problem is still a major issue in application of RNNs, it is good to see there is another work dedicating to explore its answer. - The perspective of the proposed method is interesting. Connecting temporal dynamics with momentum in SGD is a novel and reasonable assumption. And the paper gives a good comparison of similarities between hidden states and SGD steps.

Weaknesses: - Although the paper puts an emphasis on the relations between SGD and temporal hidden states, I find integrated momentum is quite similar to forget/input gating mechanism in form, which helps update hidden states by controlling forgetting/memorizing information in conventional LSTM. The only difference is forget/input gating is totally data-driven, while the proposed momentum is hand-crafted. So in my opinion, Momentum RNN introduces additional hyper-parameters and could be very sensitive to the parameter selection, leading to a painful tuning process and limited scalability. This may partially verified by Figure 6, which I consider as a disadvantage to conventional LSTM. - Important experiments are missing. For example, since the paper claim momentum RNN can alleviate gradient vanishing problem, it should be tested on related tasks such as copy/adding problems, which are commonly used for majority of RNN evaluations. Besides, only conventional baselines and several of its own variants are evaluated on reported tasks while other closely related baselines such as [39] [58] are missing. These previous works should also be included for comparison.

Correctness: The proposed method technically sounds. And empirical settings look valid.

Clarity: This paper is clearly written and easy to follow.

Relation to Prior Work: Several previous works are missing and needed to be compared with the proposed method. Please refer to the weakness section.

Reproducibility: Yes

Additional Feedback: --------------- After rebuttal ---------------- After reading your reviews and authors' feedback, I have similar concerns with the reviewer 1 and believe they should not be overlooked. Since authors' feedback cannot convince me, I will keep my original score unchanged.

[Author Response · NeurIPS 2020]

Thanks to the reviewers for their valuable feedback. We are encouraged by their endorsements that our MomentumRNN
framework: 1) makes a very novel and thought-provoking [**R3**] connection between RNN and optimization [**R3**,**R4**],
2) provides a thorough analysis integrating various forms of momentum into different RNN types [**R2**], and 3) leads to
decent improvements over baselines [**R1**,**R2**,**R3**]. Below we address the concerns raised by the reviewers.

[**R1**,**R4**] **Momentum cell is similar to gated update** $\mathbf{c}_t = \mathbf{f}_t \odot \mathbf{c}_{t-1} + \mathbf{i}_t \odot \mathbf{h}_t$ **of LSTMs and residual mappings.** We
respectfully disagree with this point. We believe there is a misunderstanding. The MomentumRNN cell updates as:
$\mathbf{v}_t = \mu\mathbf{v}_{t-1} + s\mathbf{W}\mathbf{x}_t; \mathbf{h}_t = \sigma(\mathbf{U}\mathbf{h}_{t-1} + \mathbf{v}_t)$, which introduces an auxiliary state $\mathbf{v}_t$, inspired by Nesterov momentum.
First, for the hidden state dynamics, the two-step update in momentum cell is equivalent to $\mathbf{h}_t = \sigma(\mathbf{U}(\mathbf{h}_{t-1} -$
$\mu\mathbf{h}_{t-2}) + \mu\sigma^{-1}(\mathbf{h}_{t-1}) + s\mathbf{W}\mathbf{x}_t)$, i.e., $\mathbf{h}_t$ directly depends on $\mathbf{h}_{t-1}$ and $\mathbf{h}_{t-2}$; in LSTM, $\mathbf{h}_t$ only directly depends on $\mathbf{h}_{t-1}$.
$\mu\sigma^{-1}(\mathbf{h}_{t-1})$ is the key term that helps alleviate the vanishing gradient. Second, $\mathbf{c}_t = \mathbf{f}_t \odot \mathbf{c}_{t-1} + \mathbf{i}_t \odot \mathbf{h}_t$ differs from
$\mathbf{v}_t = \mu\mathbf{v}_{t-1} + s\mathbf{W}\mathbf{x}_t$, where $\mathbf{f}$ and $\mathbf{i}$ are tensors while $\mu$ and $s$ are scalars; also, $\mathbf{h}_t$ is the hidden state while $\mathbf{x}_t$ is the
input. Third, $\mathbf{v}_t = \mu\mathbf{v}_{t-1} + s\mathbf{W}\mathbf{x}_t$ differs from residual mapping ($\mathbf{v}_t = \mathbf{v}_{t-1} + \mathcal{F}(\mathbf{v}_{t-1})$). As pointed out by [**R3**], "*In*
*LSTM, the cell state performs additive integration of the input so that the gradients do not vanish. However, LSTM's*
*sigmoid and tanh gates make gradients vanish. MomentumRNN performs additive integration of the input without any*
*additional gates and hence has a much better gradient flow.*" We hope reviewers can reevaluate this crucial point.

[**R1**] **Analogy between MomentumRNN and optimization methods.** We do not aim to equate RNNs with gradient
descent. Rather, we aim to bring in new ideas from optimization to design better RNNs. The nonlinear activation can be
considered as a projection, and, based on our experiments, the acceleration from momentum is preserved. Moreover, the
resulting momentum cell derived from optimization algorithms with momentum can alleviate the vanishing gradient.

[**R1**] **Analysis is w.r.t. the vanilla RNN. Also, it is not obvious that an appropriate** $\mu$ **exists.** In Sec. 2.3, we proved
that $\mu\sigma^{-1}(\mathbf{h}_{t-1})$ in the MomentumRNN cell $\mathbf{h}_t = \sigma(\mathbf{U}(\mathbf{h}_{t-1} - \mu\mathbf{h}_{t-2}) + \mu\sigma^{-1}(\mathbf{h}_{t-1}) + s\mathbf{W}\mathbf{x}_t)$ is key to alleviate the
vanishing gradient in vanilla RNNs, which generalizes to the analysis of other RNNs. Our empirical study in Fig. 2 is not
for a vanilla RNN, but for the SOTA DTRIV model (see Appendix A.4). Our experiments show that MomentumRNNs
significantly outperform all studied baseline RNN models, which empirically validates that an appropriate $\mu$ exists.

[**R1**,**R4**] **Toy datasets and lack of comparison to other SOTA RNN baselines.** We used the (P)MNIST, TIMIT, and
Penn TreeBank (PTB) benchmarks, which are not toy. Our momentum-based method can be applied to many other
RNNs, including those that that address vanishing gradients, to improve their performance. In Sec. 3 and 4, we have
shown this in the case of LSTM and expRNN (DTRIV), a SOTA RNN model. Our approach outperforms both.

[**R1**,**R3**] **More related work on long-term dependencies and saturations of LSTM.** We appreciate the suggestions
and will add JANET, NRU, and more papers on alleviating long terms dependency issues in the revision.

[**R2**] **Why the MomentumRNN provides improvements over RNN is not clear.** MomentumRNN mitigates vanish-
ing gradients as analyzed in Sec 2.3. The reason it can theoretically accelerate convergence, improve robustness, and
lead to higher performance is under our study. In particular, we are studying the continuum limit of the MomentumRNN.

[**R2**] **Compare the computation cost of MomentumRNN & RNN to reach similar acc.** When training on the
PMNIST task using 256 hidden units, we observe that *to reach 92.29% test acc. for LSTM (see Tab. 1), LSTM needs*
**699min** *while MomentumLSTM & RMSPropLSTM (our best model for this task) only need* **416min** *and* **403min***, resp*.

[**R3**] **Why should RMSPropLSTM work?** RMSPropLSTM inherits its adaptive step size from RMSProp. In training,
$\mathbf{W}\mathbf{x}_t$ is rescaled adaptively and can improve training and enhance performance for certain tasks.

[**R3**] **Why using different optimizers for different tasks? Results on using Adam for all models.** We used the
default optimizers that achieve SOTA results for different tasks, e.g., we used *RMSProp for (P)MNIST, Adam for TIMIT,*
*and SGD for PTB*. Using Adam to train LSTM, MomentumLSTM, AdamLSTM, RMSPropLSTM, and SRLSTM with
256 hidden units on PMNIST, we obtain test accuracy of $92.05 \pm 0.63\%$, $92.47 \pm 0.35\%$, $92.53 \pm 0.26\%$, $93.86 \pm 0.24\%$
and $92.17 \pm 1.37\%$, resp. Adam trainings for DTRIV and MomentumDTRIV with 512 hidden units on PMNIST yield
the test accuracy of $95.78 \pm 0.21\%$ and $96.01 \pm 0.10\%$, resp. Adam training yields worse results than SGD on PTB.

[**R3**] **1) Why using only the MomentumRNN in language modelling task? 2) Fig. 6 shows that often momen-**
**tum=0 is the best thing to do? 3) Why is forget gate initialized to -4? 1)** We compare 3-layer MomentumLSTM
with 3-layer LSTM in our paper (see Tab. 3). The test PPL for AdamLSTM, RMSPropLSTM, and SRLSTM are
$61.11 \pm 0.31$, $64.53 \pm 0.20$, and $58.83 \pm 0.62$, resp. **2)** In Fig. 6, momentum=0 yields the best results only for language
modeling task. We believe that if we do finer-scale search for momentum and step size, we can obtain a better result with
a non-zero momentum. **3)** For the TIMIT task, initializing the forget gate bias to -4 is suggested in [arXiv:1707.09520].

[**R2**,**R4**] **Additional hyper-parameters & their sensitivity.** We search $\mu$ & $s$ independently for a few options which
is not very sensitive and does not increase the computational cost much. Also, trainable parameters is under our study.

[**R4**] **Copy/adding tasks.** In copying (adding) task for sequences of length 2K (750), our MomentumLSTM, AdamL-
STM, RMSPropLSTM, and SRLSTM achieve the final training loss of 0.009 (0.162), **0.004** (0.006), 0.008 (**0.004**), and 0.01
(0.166), resp. while the training loss of the baseline LSTM is 0.01 (0.162). AdamLSTM and RMSPropLSTM converge
faster than LSTM in both tasks. Moreover, AdamDTRIV and RMSPropDTRIV converge to the same final training loss
remarkably faster than DTRIV in both tasks. Detailed results are included in the revision.

[Meta-Review · NeurIPS 2020]

This paper makes a nice connection between standard ways of regularizing the dynamics of SGD and that of RNN. Although there are some disagreements between reviewers regarding the theoretical justification, the contribution is of interest to NeurIPS audience.